# Molecular Methods for the Simultaneous Detection of Tomato Fruit Blotch Virus and Identification of Tomato Russet Mite, a New Potential Virus–Vector System Threatening Solanaceous Crops Worldwide

**DOI:** 10.3390/v16050806

**Published:** 2024-05-18

**Authors:** Marta Luigi, Antonio Tiberini, Anna Taglienti, Sabrina Bertin, Immacolata Dragone, Anna Sybilska, Franca Tarchi, Donatella Goggioli, Mariusz Lewandowski, Sauro Simoni, Francesco Faggioli

**Affiliations:** 1Council for Agricultural Research and Economics, Research Centre for Plant Protection and Certification, Via C.G. Bertero 22, 00156 Rome, Italy; 2Department of Plant Protection, Warsaw University of Life Sciences, Nowoursynowska St. 159, 02-776 Warsaw, Poland; 3Council for Agricultural Research and Economics, Research Centre for Plant Protection and Certification, Via Lanciola 12/a, 50125 Firenze, Italy

**Keywords:** *Blunervirus solani*, ToFBV, *Aculops lycopersici* Massee, TRM, tomato, eryophids

## Abstract

Tomato fruit blotch virus (ToFBV) (*Blunervirus solani*, family *Kitaviridae*) was firstly identified in Italy in 2018 in tomato plants that showed the uneven, blotchy ripening and dimpling of fruits. Subsequent High-Throughput Sequencing (HTS) analysis allowed ToFBV to be identified in samples collected in Australia, Brazil, and several European countries, and its presence in tomato crops was dated back to 2012. In 2023, the virus was found to be associated with two outbreaks in Italy and Belgium, and it was included in the EPPO Alert list as a potential new threat for tomato fruit production. Many epidemiologic features of ToFBV need to be still clarified, including transmission. *Aculops lycopersici* Massee (Acariformes: Eriophyoidea), the tomato russet mite (TRM), is a likely candidate vector, since high population densities were found in most of the ToFBV-infected tomato cultivations worldwide. Real-time RT-PCR tests for ToFBV detection and TRM identification were developed, also as a duplex assay. The optimized tests were then transferred to an RT-ddPCR assay and validated according to the EPPO Standard PM 7/98 (5). Such sensitive, reliable, and validated tests provide an important diagnostic tool in view of the probable threat posed by this virus–vector system to solanaceous crops worldwide and can contribute to epidemiological studies by simplifying the efficiency of research. To our knowledge, these are the first molecular methods developed for the simultaneous detection and identification of ToFBV and TRM.

## 1. Introduction

Tomato fruit blotch virus (ToFBV) (*Blunervirus solani* (genus *Blunervirus*, family *Kitaviridae*)) is the name for a recently identified virus species reported to infect tomato (*Solanum lycopersicum* L.). ToFBV was firstly reported in 2018 [1] at two different time points (June and September) in a limited tomato-production area in the Latium region (Central Italy). It was identified on fruit samples (*cv*. Tarquito) showing uneven, blotchy ripening and dimpling. Despite the clear symptomatology related to a viral infection, the identification of a putative etiological agent was only possible through the use of the High-Throughput Sequencing (HTS) approach, as both bioassays and molecular detection methods previously failed. The genome of this ToFBV isolate has four polyadenylated ss (+) RNA segments, each containing at least one putative Open Reading Frame (ORF) [1]. As other blunerviruses, no specific coat protein domain was found in any of the viral proteins [2]. Recently, ICTV approved the proposal to establish a new family named *Kitaviridae* containing three genera (*Blunervirus*, *Cilevirus, Higrevirus*) (https://ictv.global/taxonomy, accessed 29 February 2024 12:02 GMT; refs. [3,4]).

This family, recently recognized by the international committee of Taxonomy in 2019 [2], is composed of three genera, *Cilevirus*, *Higrevirus,* and *Blunervirus*, including viruses with multiple positive-sense, single-stranded RNA genomic segments [5]. The phylogenetic analysis of the ToFBV RNA2 ORF coding sequence for RdRp allows us to consider this virus as new species within the *Blunervirus* genus together with the two classified blunerviruses, tea plant necrotic ring blotch virus (TPNRBV, *Blunervirus camelliae*) and blueberry necrotic ring blotch virus (BNRBV, *Blunervirus vaccinii*). In 2019, ToFBV was further identified on tomato fruits (*cv*. Eshkol) showing analogous symptomatology and collected in an Italian production area close to the site of the first record [1]. The virus detection was performed using a preliminarily developed SYBR green real-time reverse-transcription polymerase chain reaction (RT-PCR) and an endpoint RT-PCR followed by Sanger sequencing [1]. Following these first identifications, ToFBV was reported in other European/non-European countries by analysing samples of different tomato cultivars from fields or stored collections; to date, ToFBV has been reported in Australia (samples collected in in 2020; ref. [1]), Brazil (samples collected in 2019; ref. [6]), Slovenia (samples collected in 2019; ref. [7]), Portugal (stored samples collected in 2015; ref. [8]), Spain (stored samples collected from 2016 to 2019; ref. [8]), Switzerland (samples collected in 2022; ref. [9]), Greece (samples collected in 2022 and 2023; ref. [10]), and Belgium (samples collected in 2022; GenBank accession numbers: PP297128.1, PP297129.1, PP297130.1, PP297131.1). In Italy, ToFBV was further reported in tomato samples collected in Sicily in 2023 [11]; in addition, it was also found in stored samples from production areas close to the 2018 and 2019 outbreaks, dating its presence in this country back to 2012 [1]. In all cases, the virus identification was performed by means of HTS, allowing all four RNAs composing the ToFBV genome to be obtained (10 complete datasets); in 2023, a further complete genome dataset from samples of *Solanum tuberosum* collected in Tunisia in 2013 was published (https://www.ncbi.nlm.nih.gov, accessed 29 February 2024 12:08 GMT). Preliminary studies reported low genome variability within the ToFBV species, except for the Australian isolate that showed a deletion of six aminoacids in movement protein [1]. As tomato is an economically important crop, grown indoors and/or outdoors across the EPPO region [12], ToFBV was recently included in the EPPO Alert list (https://www.eppo.int/ACTIVITIES/plant_quarantine/alert_list, accessed 29 February 2024 12:20 GMT). Despite many aspects of the biology, epidemiology, geographical distribution, host range, and economic impact of ToFBV being unknown, the emergence of a new virus affecting this crop potentially represents a serious threat to tomato fruit production. Recently, an ultrastructural investigation on pericarp tissues of blotched areas of ToFBV-infected fruits showed the presence of bacilliform virus-like particles of ca. 25 nm diameter and ca. 100 nm length accumulated in the perinuclear space and in the lumen of the endoplasmic reticulum; significant alterations in the pericarp cells, indicating a peculiar cytopathic effect, have been also observed [13]. Symmetry in cytopathology agrees with the phylogenetical similarity pattern in the *Kitaviridae* family.

The transmission of ToFBV is still to be elucidated. Other virus genera within the same family (*Higrevirus* and *Cilevirus* species members, i.e., hibiscus green spot virus 2 (*Higrevirus waimanalo*) and citrus leprosis virus C (*Cilevirus leprosis*), respectively) showed a common transmission pathway through false mites (*Brevipalpus* spp.) [2]; nonetheless, the other two classified blunerviruses are suggested to be transmitted by mites in the *Eriophyidae* family [14,15]. *Aculops lycopersici* Massee (Acariformes: Eriophyoidea), the tomato russet mite (TRM), seems to be a likely candidate ToFBV vector, since high population densities of TRM, worldwide, were found in most of the ToFBV-infected tomato cultivations ([6,10]; S. Simoni, personal communication). Eriophyids are phytophagous mites and feed and reproduce on all plant parts with the exception of roots. Numerous eriophyid species have economic significance due to their primary damaging action [16]; however, in the last decade, more and more virus species have been characterized as eriophyid-transmitted viruses (ETVs). ETVs and suspected ETVs show a wide variety of genomes and expression strategies [17]. ETVs are highly specialized and are only transmitted by a single species of mite [18], while a single eriophyid species can transmit more than one virus belonging to the same or a different genus and family and characterized by different morphology [17]. 

TRM is a vagrant eriophyoid mite and, differently from most eriophyoid species, is regarded as a pest in a variety of crops belonging to the Solanaceae family [19]. Presently, TRM can be found in both tropical and temperate parts of the globe with great economic impact, even though the most corroborated hypothesis may lead to a South American origin of TRM [20]. The availability of appropriate diagnostic tests for ToFBV detection is crucial given the growing number of reports and interest in the virus, particularly in light of the rising alert. Analogously, a quick and sensitive identification test, alternative to the morphologic method of TRM detection, will simplify and enhance the efficiency of research. The accurate identification of eriophyid mites at the species level requires the observation of peculiar fine morphological details, which are particularly hard to identify in view of the small dimensions of each single specimen. Such issues make morphology-based identification a laborious and time-consuming task, also requiring extensive skills and expertise. 

In this work, we reported the development and validation of two diagnostic tests, based on real-time and droplet digital amplification, for the detection and identification of ToFBV and its likely candidate vector, TRM, respectively. The use of droplet digital RT-PCR could represent an important tool for the absolute quantification of target RNAs of both virus and TRM. The real-time RT-PCR test was also developed as a duplex detection assay for both virus and vector, starting from single nucleic acid extraction and allowing the amplification of both RNA targets in a unique analysis. The validation of tests was conducted according to EPPO Standard PM 7/98 (5) [21]. To our knowledge, this is the first molecular method developed for the simultaneous detection and identification of ToFBV and TRM, a likely useful tool/approach in view of the probable threat posed by this virus–vector system to solanaceous crops worldwide.

## 2. Materials and Methods

### 2.1. Plant and Mite Samples

The reverse transcription real-time PCR (real-time RT-PCR) and droplet digital RT-PCR (ddRT-PCR) tests for ToFBV detection in plant tissues were developed and validated by analysing the samples listed in Table 1. 

The panel of samples included solanaceous leaves, roots, and fruits infected by ToFBV; healthy samples of solanaceous and other non-host plants; and samples infected by non-target viruses and viroids. These plant samples were originated from field surveys carried out in Italy (Latium region, Central Italy; Apulia region, Southern Italy) in 2022 and 2023 and from the Virus Collection (infected dried leaf tissues) available at the Council for Agricultural Research and Economics, Research Centre for Plant Protection and Certification (CREA-DC) (Rome, Italy).

The development and validation of both real-time RT-PCR and ddRT-PCR tests for the identification of TRM at species level were based on the samples listed in Table 2. The panel samples included specimens of TRM from field collections and from the rearing available at CREA-DC. The analytical specificity of the two assays was tested on specimens belonging to two other eriophyid species, *Aceria tosichella* Keifer and *Phyllocoptes adalius* Keifer, from the rearing available at the Department of Plant Protection, Warsaw University of Life Sciences, and on insect specimens belonging to the *Bemisia tabaci* (Hemiptera: Aleyrodidae) species from the rearing available at CREA-DC. 

### 2.2. Total RNA Extraction

Total RNA (TRNA) was extracted from healthy, ToFBV-infected, and mite-infested leaves, fruits, and roots of tomato and from mite and insect samples. TRNA was used for the development and validation of real-time RT-PCR (in single or duplex analysis) and ddRT-PCR assays. After a brief wash to remove soil residues, leaf, fruit, and root tissues were ground in phosphate buffer (PO_4_) 0.1 M, pH 7.2 (1:5 w:v); 100 µL of the obtained solution was used for TRNA extraction using the RNeasy plant mini kit (Qiagen, Hilden, Germany) and following the manufacturer’s instructions. For the mite and insect samples, 5 µL of RNase-free water was added to the vials containing mites; the vials were incubated 10 min at 99 °C, and after a spin down, the liquid phase was used for TRNA extraction. TRNA was extracted using the RNeasy plant mini kit following the manufacturer’s instructions, and the final elution was performed in 20 µL of molecular-grade water.

### 2.3. Primer and Probe Design

For ToFBV detection, primers and TaqMan^®^ probes were designed using all the sequences of the ToFBV putative coat protein (CP) gene available in GenBank (ORF 3 of RNA 3) [1]. For the molecular identification of TRM, primers and TaqMan^®^ probes were designed on the available sequences of the D2 region of the 28S rDNA. In both cases, the sequences retrieved in GenBank were aligned using Mega 11 software [22]. The conserved genomic regions among virus isolates or TRM specimens were selected to construct primers and probes using the PrimerExpress 3 tool (Thermo Fischer Scientific) and applying the selection criteria suggested by the ddPCR Application Guide (Biorad, bulletin 6407). Primer and probe sequences were manually adjusted when needed. Several potential primers and TaqMan^®^ probes were then tested for their inclusivity and exclusivity in silico using the Blast tool from the National Center for Biotechnology Information (NCBI, https://blast.ncbi.nlm.nih.gov/Blast.cgi accessed 29 February 2024 12:30 GMT). The primers and probes listed in Table 3 were finally selected. All the selected TaqMan^®^ probe and primer sets were synthesized by Eurofins genomics (Köln, Germany).

### 2.4. Real-Time RT-PCR Optimization for Virus Detection

Preliminary assays were performed to set up the real-time RT-PCR conditions for ToFBV detection in plant samples by including one negative and two positive leaf tomato samples (Table 1). The samples were tested at eight annealing temperatures ranging from 55 °C to 62 °C and at different primer (100, 300, 600, and 900 nM) and probe (100 and 250 nM) concentrations. 

Once the amplification conditions were optimized, reactions were carried out in 10 µL of reaction volume containing 5 µL of 2 × Mastermix, 0.25 µL of 40× RT Enzyme (both from TaqMan^®^ RNA-to-Ct™ 1-Step Kit, Thermo Fisher Scientific, Milan, Italy), 0.3 µM of each primer, 0.25 µM of labelled TaqMan^®^ probe, and 1 µL of TRNA template. The optimized one-step real-time RT-PCR cycling conditions included an RT step at 48 °C for 30 min, an initial denaturation step at 95 °C for 10 min, and 40 cycles of denaturation and annealing/elongation steps at 95 °C for 15 s and 60 °C for 1 min, respectively. Analyses were performed using the thermocycler Bio-Rad CFX 96 Touch Real-Time PCR Detection System.

### 2.5. Real-Time RT-PCR Optimization for Mite Identification

Preliminary assays were performed to set up the real-time RT-PCR conditions for the identification of TRM at the species level by including two batches of 5 and 10 TRM specimens (Table 2). The samples were tested at eight annealing temperatures ranging from 55 °C to 62 °C and at different primer (150, 450, and 900 nM) and probe (100, 250 nM and 500 nM) concentrations. 

Once the amplification conditions were optimized, reactions were carried out in 10 µL of reaction volume containing 5 µL of 2× Mastermix, 0.25 µL of 40× RT Enzyme (both from TaqMan^®^ RNA-to-Ct™ 1-Step Kit, Thermo Fisher Scientific), 0.45 µM of each primer, 0.1 µM of labelled TaqMan^®^ probe, and 1 µL of template RNA. The optimized one-step real-time RT-PCR cycling conditions included an RT step at 48 °C for 30 min, an initial denaturation step at 95 °C for 10 min, and 40 cycles of denaturation and annealing/elongation steps at 95 °C for 15 s and 62 °C for 1 min, respectively. Analyses were performed using the thermocycler Bio-Rad CFX 96 Touch Real-Time PCR Detection System.

### 2.6. Duplex Real-Time RT-PCR Optimization for Simultaneous Virus Detection and Mite Identification

One negative tomato leaf sample, two batches of 5 and 10 TRM specimens, and two leaf samples both infected by ToFBV and infested by TRM were used to set up the duplex real-time RT-PCR (Table 1 and Table 2). The samples were tested at eight annealing temperatures ranging from 55 °C to 62 °C using the primer and probe concentrations of the real-time RT-PCR. Once the amplification conditions were optimized, reactions were carried out in 10 µL of reaction volume containing 5 µL of 2× Mastermix (TaqMan™ Universal PCR Master Mix, no AmpErase™ UNG, Thermo Fisher Scientific), 0.45 µM of each primer specific for TRM, 0.1 µM of labelled TaqMan^®^ probe specific for TRM, 0.3 µM of each primer specific for ToFBV, 0.25 µM of labelled TaqMan^®^ probe for specific ToFBV, and 1 µL of template RNA. The optimized duplex real-time RT-PCR cycling conditions included an RT step at 48 °C for 30 min, an initial denaturation step at 95 °C for 10 min, and 40 cycles of denaturation and annealing/elongation steps at 95 °C for 15 s and 62 °C for 1 min, respectively. Analyses were performed using the thermocycler Bio-Rad CFX 96 Touch Real-Time PCR Detection System.

### 2.7. Droplet Digital RT-PCR Optimization for Virus Detection

The ddRT-PCR assays were performed in the QX200TM Droplet Digital PCR system (Bio-Rad, Hercules, CA, USA). According to the manufacturer’s instructions, droplets were produced using a droplet generator (Bio-Rad) and about 40 μL of the resulting emulsion was transferred to a PCR plate, which was heat-sealed using a PX1TM PCR Plate Sealer (Bio-Rad), and amplification was performed in a C1000 Thermal Cycler (Bio-Rad). The same primers and probes designed for the real-time RT-PCR targeting ToBFV were used also for the ddRT-PCR assay (Table 3). Preliminary runs were made with the same samples used in the development of the real-time RT-PCR. The tested conditions included an annealing temperature ranging from 55 °C to 62 °C and different primer (450 and 900 nM) and probe (125 and 250 nM) concentrations. Once the amplification conditions were optimized, ddRT-PCR was carried out in a 20 µL reaction volume containing 10 µL of 2× digital PCR supermix for probes (No dUTP) (Bio-Rad), 0.45 µM of each primer, 0.25 µM of TaqMan^®^ probe, DTT 10 mM, 20 U Moloney Murine Leukemia Virus Reverse Transcriptase (Mu-MLV RT, Promega, Milan, Italy), and 2 μL of TRNA. The optimized one-step ddRT-PCR cycling conditions (temperature ramp rate, 2 °C/s) included an RT step at 42 °C for 1 h, an initial denaturation step at 95 °C for 10 min followed by 50 cycles of denaturation and annealing/elongation at 94 °C for 30 s and at 60 °C for 1 min, respectively, followed by the last steps at 98 °C for 10 min and 72 °C for 5 min. Data were visualized using QuantaSoft 1.7.4.0917 software (Bio-Rad).

### 2.8. Droplet Digital RT-PCR Optimization for Mite Identification

The ddRT-PCR assays were performed using the same primers and probes designed for the real-time RT-PCR targeting the D2-28S rDNA region of TRM (Table 3). Preliminary runs were made with the same samples used in the development of the real-time RT-PCR. The tested conditions included an annealing temperature ranging from 55 °C to 62 °C and different primer (450 and 900 nM) and probe (125 and 250 nM) concentrations. Once the amplification conditions were optimized, ddRT-PCR was carried out in a 20 µL reaction volume containing 10 µL of 2× digital PCR supermix for probes (No dUTP) (Biorad), 0.45 µM of each primer, 0.25 µM of TaqMan^®^ probe, DTT 10 mM, 20 U Mu-MLV, and 2 μL of TRNA. The optimized ddRT-PCR cycling conditions (temperature ramp rate, 2 °C/s) included an RT step at 42 °C for 1 h, an initial denaturation step at 95 °C for 10 min, followed by 50 cycles of denaturation and annealing/elongation at 94 °C for 30 s and at 56 °C for 1 min, respectively, followed by the last steps at 98 °C for 10 min and 72 °C for 5 min. Data were visualized using QuantaSoft 1.7.4.0917 software (Bio-Rad).

### 2.9. Validation of the Developed Tests 

The tests were validated according to the EPPO PM 7/98 (5) [21] by assessing the analytical sensitivity and specificity, selectivity, repeatability, and reproducibility.

#### 2.9.1. Analytical Sensitivity

For ToBFV detection, three infected tomato leaf samples were ten-fold serially diluted in healthy plant tissue extracts from 10^−1^ up to 10^−9^ (Table 1). The maximum dilution of the three sample extracts giving a positive result was established as limit of detection (LOD). 

For the identification of TRM, five pools of three, five, and ten individuals and ten single individuals were prepared, with a total of 25 samples. TRNA was extracted and tested and the minimum number of individuals detected was determined (Table 2). The data obtained with ddRT-PCR were also used to correlate the copies/µL to the number of specimens. 

The LOD was evaluated also for the duplex real-time RT-PCR assay by testing the same samples used to validate the analytical sensitivity of the single assays. To further assess if multiplexing could affect the analytical sensitivity of the mite amplification, a ToFBV-infected tomato leaf sample was macerated in PO_4_ buffer and diluted with healthy leaf sap to reach the LOD. Three boiled TRM specimens were added to each dilution; then, the TRNA was extracted as described above and amplified following the duplex real-time RT-PCR protocol. 

#### 2.9.2. Analytical Specificity

The inclusivity and exclusivity of the developed tests were firstly assessed in silico by questioning the NCBI database. The inclusivity of ToFBV detection assays was assessed by analysing 14 infected tomato samples collected in different years at different sites (Table 1); the inclusivity of the TRM identification assay was assessed by analysing five batches of three individuals from both rearing and field collections (Table 2). The exclusivity of the ToFBV detection assays was tested against 20 relevant tomato viruses and viroids and on healthy samples of *S. lycopersicum*, *S. nigrum,* and *Convolvulus arvensis* (Table 1). The exclusivity of the mite identification assays was assessed by analysing five batches of five individuals belonging to the eriophyid species *A. tosichella* and *P. adalius* along with one sample of *B. tabaci* (Table 2).

#### 2.9.3. Selectivity

Selectivity was assessed only for ToFBV detection assays. TRNA from tomato fruits, leaves and roots was used to determine if the matrix could affect the performance of real-time RT-PCR (both single and duplex) and ddRT-PCR tests (Table 1). 

#### 2.9.4. Repeatability and Reproducibility

The repeatability and reproducibility of the developed tests were assessed by performing six different nucleic acid extractions from the same plant/mite material and by amplifying these samples in replicates. In detail, for the real-time RT-PCR and ddRT-PCR assays targeting ToFBV, three extractions were made from an infected tomato sample that had been macerated in PO_4_ buffer and diluted with healthy leaf sap to achieve a low viral concentration (10^−5^; Table 1). Then, on a different day, the same extraction was repeated to assess the reproducibility. Moreover, the real-time assay was also performed with different equipment (AriaMx Real-time PCR System, Agilent, CA, USA), selecting ROX as a passive reference dye. 

For evaluating the two parameters of the real-time RT-PCR and ddRT-PCR for mite identification, two samples (three TRM specimens/sample) were extracted and tested on two different days (Table 2). Moreover, the real-time assay was also performed with different equipment (AriaMx Real-time PCR System, Agilent, CA, USA), selecting ROX as the passive reference dye.

For the evaluation of the duplex assay, a ToFBV-infected sample was macerated in PO_4_ buffer and diluted with healthy leaf sap to reach a low virus concentration; then, 24 already-boiled TRM were added, and the TRNA was extracted six times (three extractions/day) and tested (Table 1 and Table 2). Likewise, the test was performed with different equipment (AriaMx Real-time PCR System, Agilent, CA, USA), selecting ROX as a passive reference dye.

## 3. Results

### 3.1. Optimization of the Real-Time and Droplet Digital RT-PCR Tests for Virus Detection

The selected primers and TaqMan^®^ probe targeting the ToFBV-CP gene were shown to anneal with all the virus isolates included in the *in silico* analysis for inclusivity, suggesting they potentially amplify all the known genetic variants of ToFBV. Primers and probes were specific for ToFBV, as shown by the *in silico* exclusivity tests (Blast tool, NCBI) performed against the whole GenBank database after excluding the ToFBV sequences. After selecting the primers and probes, preliminary reactions were performed to optimize the real-time RT-PCR conditions, namely annealing temperature and primer and probe concentrations. In all the preliminary reactions, the primer/probe sets specifically detected the target, and no amplification signals were obtained from negative controls. As reported in Figure 1a, no differences were evident amongst the eight different annealing temperatures (55–62°C) tested. The annealing temperature was then fixed at 60 °C, being a versatile temperature that can be easily adapted to multiplex tests. Successively, different primer and probe concentrations were tested at the annealing temperature of 60 °C. Differences in primer concentrations did not affect the efficiency of the test, whereas the effect of the probe concentration on the relative fluorescence units (RFUs) of the reaction was considerable (Figure 1b). According to these results, the optimal primer and probe concentrations were set up at 300 nM and 250 nM, respectively.

The ddRT-PCR assay was developed using the same primers and probe designed for the real-time RT-PCR and the same step-by-step approach. First, the 55–62 °C annealing temperatures were tested, and the highest number of target copies per µL was obtained at 60 °C (Figure 2a). Concerning the primer and probe concentrations, the best separation of the positive and negative droplets and the highest number of target copies were obtained using 450 nM of each primer and 250 nM of probe (Figure 2b); therefore, this combination was established as the working concentration of the test.

### 3.2. Optimization of the Real-Time and Droplet Digital RT-PCR Tests for Mite Identification

The selected mite-specific primers and TaqMan^®^ probe were shown to anneal with all the TRM D2- 28S rDNA sequences included in the *in silico* analysis for the inclusivity. The *in silico* exclusivity analysis (Blast tool, NCBI), performed against the whole GenBank database after the exclusion of the target sequences, confirmed that these primers and probe are specific for TRM. The primers and probe were therefore selected for further analyses.

Preliminary reactions were then performed to optimize the real-time RT-PCR conditions. In all the preliminary reactions, the primer/probe set has shown to specifically amplify the target, and no amplification signals were obtained from negative controls. As reported in Figure 3a, differences in the shapes of the amplification curves were evident amongst the different temperatures tested (55–62°C), with the best performances obtained at 62 °C. Therefore, 62 °C was chosen as the annealing temperature to be used in the assays. 

The different primer and probe concentrations were tested at the annealing temperature of 62 °C. Differences in probe concentration altered the shapes of the curves, and a plateau was reached with 100 nM only. Differences in primer concentration modified the Cq of the reaction, and the concentration of 450 nM provided amplifications at the lowest Cq values; accordingly, the optimal primer and probe concentrations were set at 450 nM and 100 nM, respectively (Figure 3b).

The ddRT-PCR conditions were developed using the same primers and probe designed for the real-time PCR and using the same step-by-step approach. First, the annealing temperatures ranging from 55 °C to 62 °C were tested, and the highest number of copies/µL of the target sequence was obtained at 56 °C (Figure 4a). Successively, the concentrations of primers and probe were tested, and the combination of 450 nM of primers and 250 nM of the probe was selected according to the separation of the positive/negative droplets and the number of copies/μL (Figure 4b).

### 3.3. Optimization of the Duplex Real-Time RT-PCR

The primer and probe sets selected for ToFBV detection and TRM identification (Table 3) were combined in the same reaction mixture to develop a duplex real-time RT-PCR able to amplify both the virus and mite RNA at the same time. The same concentrations of the single assays were used, and the annealing temperature of 62 °C resulted in being optimal for the amplification of both ToFBV and TRM RNA and was therefore chosen as the annealing temperature of the duplex assays (Figure 5).

### 3.4. Validation of the Assays

#### 3.4.1. Analytical Sensitivity

Ten-fold dilutions of three ToFBV-infected tomato samples were analysed to evaluate the LOD of the real-time RT-PCR and the ddRT-PCR assays targeting the virus. The LOD resulted in being 10^−6^ for both the tests (Table 4).

Analytical sensitivity was evaluated using samples, consisting of one, three, five, and ten specimens each, in order to identify TRM. According to the results, three was the minimum number of mite specimens that could be identified by both the real-time RT-PCR and ddRT-PCR tests, and this value was thus established as the LOD (Table 5).

Analysing the data obtained from the ddRT-PCR of a specific number of TRM specimens, a clear relation between the number of individuals and the number of copies/µL was observed (Figure 6). 

For the duplex real-time RT-PCR, the obtained LOD was the same of the single assays: 10^−6^ for ToFBV and three specimens for TRM. The potential effect of virus concentration on the efficiency of the duplex assay in amplifying the mite RNA was assessed by artificially adding the minimum detectable number of mites (three specimens) to plant samples prepared with an increasing ToFBV dilution. Three TRM specimens were confirmed to be detectable with the real-time RT-PCR in all the samples, regardless of the virus concentration (Figure 7a). Nonetheless, the Cq values of TRM amplification increased proportionally to the concentration of ToFBV (Figure 7b). 

#### 3.4.2. Analytical Specificity

The inclusivity of the assays for ToFBV detection was demonstrated by analysing 14 virus isolates from two different plant species, *S. lycopersicum* and *S. nigrum*, collected in different Italian regions and in different years (Table 1). All samples tested positive both in real-time RT-PCR and in ddRT-PCR (Appendix A). For the exclusivity, the most relevant viruses and viroids affecting tomato and other solanaceous crops were tested along with TRNA from healthy samples of possible hosts of the virus (Table 1). No positive signal was obtained (Appendix A). 

Analytical specificity was also assessed for the tests targeting TRM. Also, in this case, specimens collected in different years, in different Italian regions, and from different hosts (Table 2) were successfully amplified using both real-time RT-PCR and ddRT-PCR (Appendix A). Other arthropod species were tested for exclusivity without any amplification signals. 

The same samples tested in the single assays were also used for assessing the analytical specificity of the duplex assays, and the results were completely concordant with those expected. According to these results, the inclusivity and exclusivity of all the tests were assessed as 100%. 

When tested in duplex real-time RT-PCR, six mite samples collected in open fields from two infected tomato plants (Table 2) were positive for both TRM and ToFBV. 

#### 3.4.3. Selectivity

Selectivity was only evaluated for the ToFBV detection tests by analysing different infected matrices. All the real-time RT-PCR (both single and duplex) and ddRT-PCR assays were able to amplify ToFBV in fruits, leaves, and roots even if the virus concentration varied according to the matrix. Indeed, the highest amount of virus was detected in leaf (L), followed by fruit (F), and then root (R) (Figure 8). 

#### 3.4.4. Repeatability and Reproducibility

A ToFBV-infected sample diluted with healthy leaf sap to reach a low (10^−5^) virus concentration was extracted three times on different days and then tested by both real-time RT-PCR and ddRT-PCR. Real-time RT-PCR analysis was also performed using different equipment. All the samples tested positive with comparable Cq values or n copies/µL (Figure 9a,b), confirming that all the assays for ToFBV detection were repeatable and reproducible. Concerning the real-time RT-PCR and ddRT-PCR for mite identification, two pools of three specimens of TRM were extracted and tested on two different days and with two different instruments of real-time PCR. The target was amplified in all the samples, and both the Cq values and the n copies/µL were comparable amongst the experiments (Figure 9c,d), confirming that the assays were repeatable and reproducible. 

A different approach was used to determine the repeatability and reproducibility of the duplex real-time RT-PCR test: a ToFBV-infected sample diluted until 10^−5^ was added to TRM specimens (Table 1 and Table 2), and the TRNA was repeatedly extracted and tested with two different instruments (CFX 96, Bio-Rad and AriaMx, Agilent). In all cases, both virus and mite targets were amplified, and the Cq values were comparable amongst the experiments (Figure 10). According to this, it can be stated that the duplex test was also repeatable and reproducible.

## 4. Discussion

The advancement of rapid, accurate, and sensible demonstrative techniques is one key tool to control plant infections, in order to set up appropriate phytosanitary measures for preventing the introduction and spread of pathogens and their vectors. In particular, the threat posed by emerging viruses often lies in the lack of available diagnostic tests, mainly ascribable to limited knowledge of the pathogen. Recently, ToFBV has been included in the EPPO Alert list (https://www.eppo.int/ACTIVITIES/plant_quarantine/alert_list, accessed 18 March 2024 15:50 GMT) due to multiple reports being recorded in European countries and other continents, suggesting a higher spread than previously expected [1,6,8]. Its impact on tomato fruit production is still unknown but, due to the economic importance of the crop across the EPPO region, both in greenhouses and in open fields, the emergence of a new virus causing symptoms to fruit pushes to increase the awareness of its phytosanitary risk.

Arthropod vectors represent, in many cases, a major route for the transmission of plant viruses; therefore, they are a key target for controlling plant diseases worldwide. In view of the evidence, indicating TMR as a possible candidate in transmitting ToFBV [6,10], more experiments are needed to confirm and characterize this way of transmission. Such required studies, as well as vector-monitoring activities, would be greatly supported by the availability of quick and sensitive tools simultaneously allowing virus detection and eriophyid species identification. In fact, eriophyid identification is mainly performed by morphological observation, a time-consuming process requiring highly skilled staff. The main issues in the morphological identification of eriophyid mites are their tiny size, similarity among different species of distinguishing traits, and tendency towards a hidden lifestyle [23,24,25]. In particular, for TRM, early symptoms (i.e., chlorosis on leaves, shades on the stem) can be easily overlooked, while more severe symptoms, like stem and leaf browning, may be in turn misdiagnosed as fungi infections [26]. Moreover, due to the high reproducing rate of TRM, they appear generally detectable at high population densities, impairing effective control [27,28]. 

Over the last three decades, extensive research has been conducted to develop easier tests for eriophyid mites’ detection and quantification in order to better estimate population density in sampling: various methods were proposed and reported in the literature, including, for example, portable gas chromatography [29], sticky tape imprinting, and chlorophyll fluorescence measurements [30]. All these methods proved to be more straightforward than morphological observation, but still need labour-intensive or particular light conditions, making them unsuitable for large-scale monitoring. For the latter purpose, an effective identification tool, both for viruses and mites, appropriate for non-specialist operators is required for application at borders and in activities of monitoring and research. More recently, the molecular approach to such identification with the above-mentioned features of efficiency and user-friendliness has been explored for insects and other mites [31,32,33]. The mitochondrial gene coding for the cytochrome-c oxidase I (COI) has been chosen as a reference point for a universal system of animal species identification (DNA barcoding; [34,35]), and it is widely used as a versatile tool for species identification within most of the arthropod orders. As well, ribosomal DNA has repeatedly provided insights into the genetic basis of evolution of recently diverged arthropod species and genera, and it is often a good candidate as a molecular taxonomic key for species discrimination [36,37]. For eriophyid species, both the ribosomal ITS and the D1–D12 domains of the 28S region have proven to be informative [20,38,39,40]. In addition, in the case of candidate ETVs, a diagnostic pipeline using RT-PCR was developed and proved to be effective for the simultaneous identification of eriophyid at the species level and the detection of viruses [41]; in fact, by the incorporation of a reverse transcription step, not only single-copy genomic DNA but also multiple transcripts are targeted, increasing the total amount of template, thus decreasing the minimum number of individuals to be detected; in the meantime, it also allows the amplification of RNA viruses. Concerning TRM, the sequenced genome provided a resource for the development of molecular tests and simplified the search for species-specific target DNA. 

In this context, in order to fill the existing diagnostic gap and facilitate ongoing studies on the biology and epidemiology of the virus–vector system, molecular tests based on real-time RT-PCR were developed to concurrently detect and identify ToFBV and TRM, in single or duplex assays. The optimized real-time RT-PCRs targeting both ToFBV and TRM were then transferred to the RT-ddPCR assay. The developed tests were validated following EPPO PM 7/98 (5) guidelines [21]. Such validation is a prerequisite for including a test in a regime of official controls that could be required by National Plant Protection Organizations (NPPOs), triggered after the inclusion of ToFBV in the EPPO alert list. The ToFBV assay developed in this study targeted the putative CP gene (ORF3-RNA3) and provided sufficient variability to detect all available isolates of the target virus. The RT-PCR system for the identification of TRM at the species level were designed on the variable region D2 of the 28S ribosomal DNA, which already proved to be species-specific among eriophyids [41]. 

The analytical sensitivity obtained for ToFBV in real-time RT-PCR and ddPCR assays was in accordance with those of tests for the detection of other viruses in tomato with the same methods [42,43]. In addition, when the real-time test was performed in a duplex assay also detecting TRM, analytical sensitivity was not affected compared to the single tests, although some increases in the Cq values of the TRM amplification curves were observed at high ToFBV concentrations. In addition, both single and duplex assays showed optimal analytical specificity in both inclusivity and exclusivity, ruling out the chance of missing divergent isolates of the virus and/or non-target amplifications. Moreover, for the real-time tests, both single and duplex assays proved to be reproducible, as well as with a different instrument, and repeatable. As for the single assay, EPPO PM 7/98 (5) guidelines in molecular methods for both virology and acarology were followed [21]; for the duplex assay, a wider approach was also used for testing the repeatability and reproducibility of the whole diagnostic procedure, including nucleic acid extraction, which is often a critical step in the molecular identification of mites. By this approach, the performance of the test in different extractions of the same ToFBV sap at low concentrations spiked with a fixed number (close to the minimum number of individuals detected) of TRM specimens was evaluated; notably, neither extraction nor amplification affected the repeatability and reproducibility when performed in different events and on different days, respectively. 

The duplex real-time assay demonstrated its effectiveness when tested on field samples, possibly supporting further research on the biology and epidemiology of this virus–vector-associated system, in order for it to be completely understood and to assess the actual risk to tomato crops worldwide. The ability of the test both to detect ToFBV and identify TRM in samples collected in open fields from infected plants will greatly support acquisition and transmission laboratory experiments, mainstreaming the unambiguous identification of the vector. 

The real-time PCR assay of TRM was performed with just three specimens from a sample, the minimum number of individuals for the species identification. Such performance will greatly enhance the accuracy of the estimation of population densities in collected samples: TRM populations are typically underestimated in surveys [44], and this issue impairs the correct and appropriate management of the pest and, the timely establishment of required phytosanitary measures. During validation, the test also proved to be highly specific, succeeding in correctly identifying target populations collected both in different areas and on different hosts, thus covering potential genetic diversity; most notably, it did not detect any of the non-target mites associated with the tomato host which were tested in an exclusivity assessment. Such high levels of sensitivity and specificity, together with the 100% repeatability and reproducibility obtained during validation, make this test a powerful tool for mite identification by staff not owing to the highly specialized expertise on eriophyid morphology, for example, in phytosanitary facilities at borders facing the risk of introducing pests to/from their countries. All the abovementioned advantages indicate the feasibility of the routine use of this test for the detection of TRM.

The validation of the ddRT-PCR test on both ToFBV and TRM indicated the same levels as the real-time RT-PCR for all performance criteria. The added value of this test, which requires advanced and, in any case, costly equipment, is the chance to directly estimate the copies/ul of the target. For ToFBV, this allows a measure of viral copies (coded by the putative CP) coded by genomic and subgenomic RNA to be provided without a standard curve as a calibrator, avoiding the need to perform several additional steps such as in vitro transcription assays, required in the case of RNA viruses. In the ddRT-PCR amplifying TRM, a linear relationship was observed between the number of specimens and the concentration of target RNA in copies/ul. This correlation could be used to ascertain the infestation level of samples with even higher accuracy than real-time RT-PCR and could be helpful in future surveys assessing the prevalence of vectors in tomato crops in both affected and virus-free areas. 

The importance of molecular approaches targeting both virus and mite vectors was already proven for other economically important virus–eriophyid associations, such as rose rosette emaravirus and *Phyllocoptes fructiphilus* [41]. The present study provided both single and duplex assays for both ToFBV detection and TRM identification at the species level. These assays fully satisfy the international requirements of validation for plant pest diagnosis (EPPO PM 7/98 (5), ref. [21]). Such a sensitive, reliable, and validated test represents an important diagnostic tool for quarantine purposes and routine monitoring in the field, as well as for epidemiological studies.

## Figures and Tables

**Figure 1 viruses-16-00806-f001:**
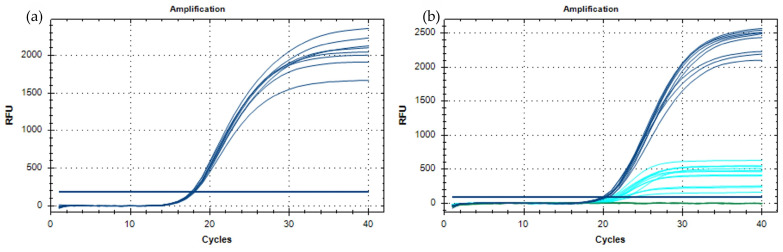
Optimization of the real-time RT-PCR for ToBFV detection. (**a**) Amplification curves obtained at eight annealing temperatures ranging from 55 °C to 62 °C. (**b**) Amplification curves obtained with different primer and probe concentrations: 250 nM probe (blue); 100 nM probe (light blue). The blue horizontal line represents the baseline, the green one the negative control samples.

**Figure 2 viruses-16-00806-f002:**
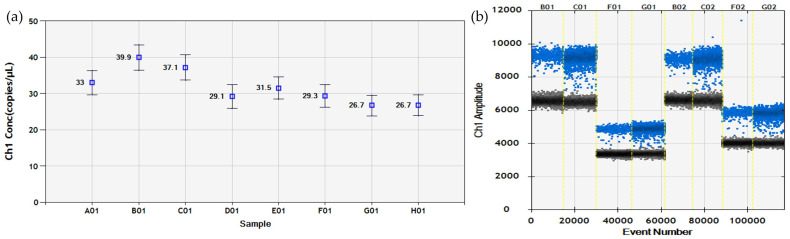
Optimization of the ddRT-PCR for ToBFV detection. (**a**) The number of copies/µL obtained by amplifying the same sample at eight annealing temperatures (A01-62 °C; B01-61.5 °C; C01-60.6 °C; D01-59.3 °C; E01-57.7 °C; F01-56.4 °C; G01-55.5 °C; H01-55.0 °C). The highest concentrations (B01 and C01) were obtained at about 60 °C. (**b**) Amplification results of two samples obtained with different primer and probe concentrations: 450 nM + 250 nM (B01 and C01); 450 nM + 125 nM (F01 and G01); 900 nM + 500 nM (B2 + C2); 900 nM + 125 nM (F02 + G02). Negative droplets (black), FAM-positive droplets (blue).

**Figure 3 viruses-16-00806-f003:**
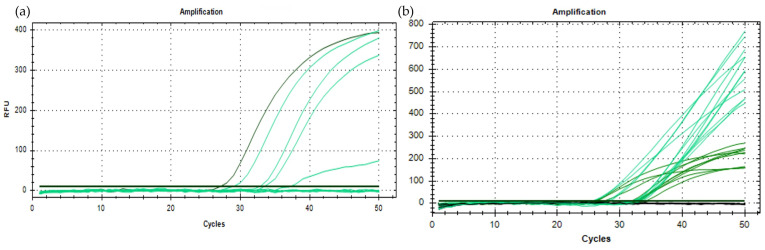
Optimization of the real-time PCR for the identification of TRM at the species level. (**a**) Amplification curves obtained at eight annealing temperatures ranging from 55 °C to 62 °C. The best curves obtained at 62 °C are shown in green; the curves obtained with the other temperatures are shown in light green. (**b**) Amplification curves obtained with different primer and probe concentrations: 100 nM probe (green); 250 and 500 nM probe (light green). The curves with a Cq value < 30 were obtained with 450 nM primers, and the curves with a Cq value > 30 were obtained with 100 and 900 nM primers. The green horizontal line represents the baseline.

**Figure 4 viruses-16-00806-f004:**
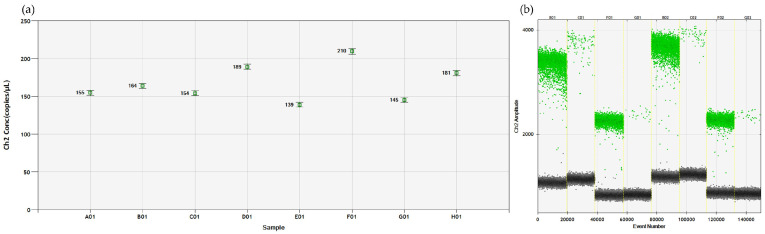
Optimization of the ddPCR for the identification of TRM at species level. (**a**) The number of copies/µL obtained by amplifying the same sample at eight annealing temperatures (A01-62 °C; B01-61.5 °C; C01-60.6 °C; D01-59.3 °C; E01-57.7 °C; F01-56.4 °C; G01-55.5 °C; H01-55.0 °C). The highest concentration, F01, was obtained at 56 °C. (**b**) Amplification results of two samples obtained with different primer and probe concentrations: 450 nM + 250 nM (B01 and C01); 450 nM + 125 nM (F01 and G01); 900 nM + 500 nM (B2 + C2); 900 nM + 125 nM (F02 + G02). Negative droplets (black), HEX-positive droplets (green).

**Figure 5 viruses-16-00806-f005:**
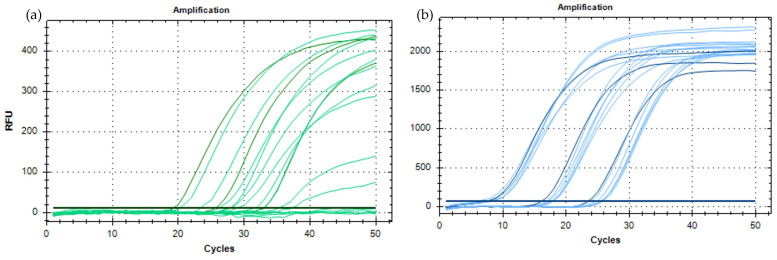
Optimization of the duplex real-time RT-PCR performed at eight annealing temperatures ranging from 55 °C to 62 °C. (**a**) Amplification curves obtained for TRM (green); curves obtained at 62 °C (dark green) are representative of the optimal annealing temperature. (**b**) Amplification curves obtained for ToFBV (blue); curves obtained at 62 °C (dark blue) are representative of the optimal annealing temperature. The green and blue horizontal lines represent the baselines.

**Figure 6 viruses-16-00806-f006:**
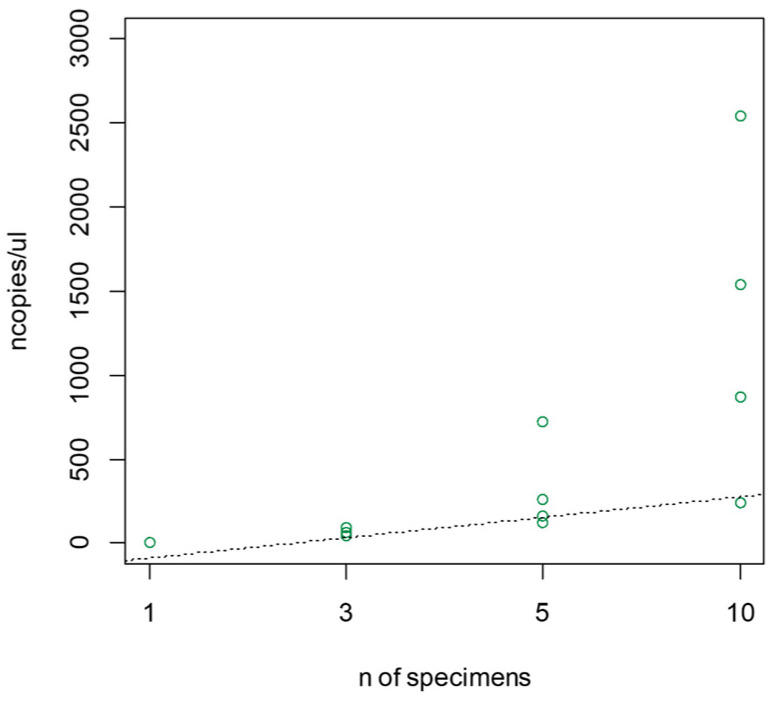
Results of the amplification obtained from different numbers of TRM specimens (1, 3, 5, 10) reported as copies/µL. The dotted line represents the interpolating line (slop e = 121.01; intercept = −208.9; R^2^ = 0.96.

**Figure 7 viruses-16-00806-f007:**
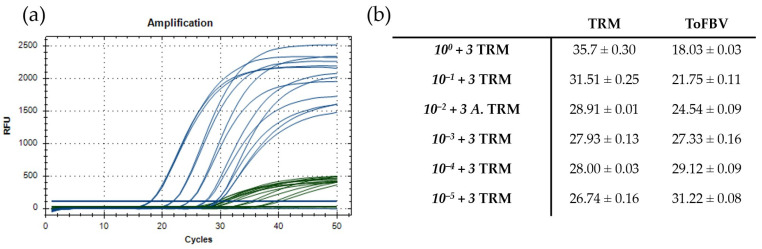
Results of the assays on mixed infected samples (ToFBV+ 3 specimens of TRM). (**a**) Results of the duplex real-time RT-PCR. Amplification of ToFBV (blue); amplification of TRM. (green). (**b**) Table reporting the Cq values and the standard deviation of the technical replicates for the duplex real-time RT-PCR. The green and blue horizontal lines represent the baselines.

**Figure 8 viruses-16-00806-f008:**
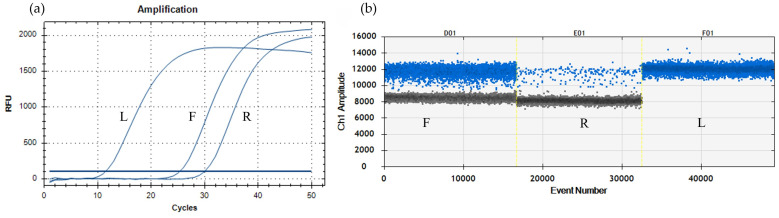
Selectivity assays on different matrixes from the tomato plant (sample 95—Table 1): L— leaf; F—fruit; R—root. (**a**) Results of the duplex real-time RT-PCR. (**b**) Results of the duplex ddRT-PCR. Negative droplets (black), FAM-positive droplets (blue). The blue horizontal line represents the baseline.

**Figure 9 viruses-16-00806-f009:**
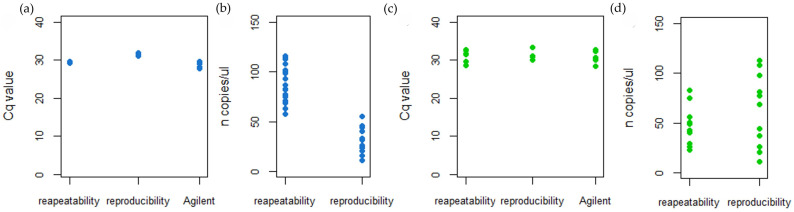
Repeatability and reproducibility of real-time RT-PCR and ddRT-PCR assays for ToFBV detection. (**a**) Cq values obtained in real-time RT-PCR, testing the same plant material at a low viral concentration. The plant material was extracted six times, and each sample was amplified in three technical replicates, on two different days, and using two different types of equipment; (**b**) n copies/µL obtained in ddRT-PCR, testing the same six extracts in three technical replicates on two different days. The repeatability and reproducibility of real-time RT-PCR and ddRT-PCR assays for the dentification of TRM (**c**) Cq values obtained in real-time RT-PCR, testing the same two samples at the minimum number of mites detected, in three technical replicates, on two different days and using two different types of equipment; (**d**) n copies/µL obtained in ddRT-PCR, testing the same mite samples in three technical replicates on two different days.

**Figure 10 viruses-16-00806-f010:**
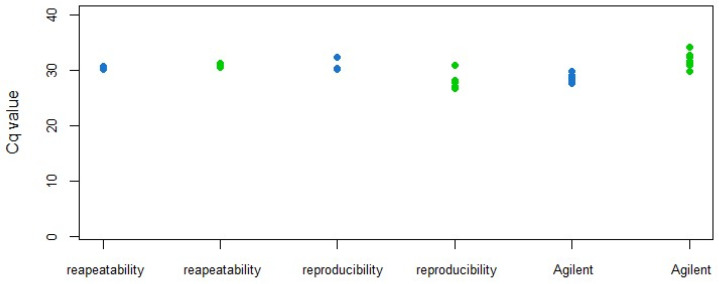
Repeatability and reproducibility evaluation of the duplex real-time RT-PCR; results for ToFBV detection and TRM are reported in blue and green, respectively. Cq values obtained in duplex real-time RT-PCR, testing the same sample at low viral concentration, added with 24 boiled mites, extracted six times and amplified in three technical replicates, on two different days using two different types of equipment (BIORAD CFX 96 and AriaMx by Agilent).

**Table 1 viruses-16-00806-t001:** List of the plant samples used (X) in the set-up and validation of the real-time RT-PCR and ddRT-PCR assays for ToFBV detection. Some samples were extracted and tested as leaves (L), roots (R), and fruits (F).

Plant Samples				Test Set-Up	Performance Criteria of Validation (EPPO PM7/98(5))				
Phytosanitary Status	N° of Tested Samples	Species (Sample ID)	Matrix		Analytical Sensitivity	Analytical Specificity/Inclusivity	Analytical Specificity/ Exclusivity	Selectivity	Repeatability and Reproducibility
**Healthy**	1	*S. lycopersicum*	L	X	X	X			
**ToFBV**	1	*S. lycopersicum* (92) ^a^	L			X			
**ToFBV**	1	*S. lycopersicum* (94) ^a^	R, L, F			X		X	
**ToFBV**	1	*S. lycopersicum* (95) ^a^	R, L, F	X	X	X		X	
**ToFBV**	1	*S. lycopersicum* (96) ^a^	F			X			
**ToFBV**	1	*S. lycopersicum* (97) ^a^	F			X			
**ToFBV**	1	*S. lycopersicum* (98) ^a^	R, L, F			X		X	
**ToFBV**	1	*S. lycopersicum* (99) ^a^	R, L, F	X	X	X		X	X
**ToFBV**	1	*Solanum nigrum* (1) ^b^	L		X	X			
**ToFBV**	1	*S. nigrum* (2) ^b^	L			X			
**ToFBV**	1	*S. lycopersicum* (4) ^b^	L			X			
**ToFBV**	1	*S. lycopersicum* (5) ^b^	L			X			
**ToFBV**	1	*S. lycopersicum* (104) ^b^	L			X			
**ToFBV**	1	*S. lycopersicum* (1B) ^c^	L			X			
**ToFBV**	1	*S. lycopersicum* (1C) ^c^	L			X			
**Healthy**	4	*S. lycopersicum*	L				X		
**Healthy**	5	*S. nigrum*	L				X		
**Healthy**	3	*Convolvolus arvensis*	L				X		
**CEVd**	1	*S. lycopersicum* ^d^	L				X		
**CLVd**	1	*S. lycopersicum* ^d^	L				X		
**CMV**	1	*S. lycopersicum* ^d^	L				X		
**INSV**	1	*S. lycopersicum* ^d^	L				X		
**PCFVd**	1	*S. lycopersicum* ^d^	L				X		
**PePMV**	1	*S. lycopersicum* ^d^	L				X		
**PmoV**	2	*S. lycopersicum* ^d^	L				X		
**PSTVd**	1	*S. lycopersicum* ^d^	L				X		
**PVY**	2	*S. lycopersicum* ^d^	L				X		
**TASVd**	1	*S. lycopersicum* ^d^	L				X		
**TCDVd**	1	*S. lycopersicum* ^d^	L				X		
**TICV**	2	*S. lycopersicum* ^d^	L				X		
**TMV**	1	*S. lycopersicum* ^d^	L				X		
**ToBRFV**	1	*S. lycopersicum* ^d^	L				X		
**ToCV**	2	*S. lycopersicum* ^d^	L				X		
**ToLCNDV**	1	*Cucurbita pepo* ^d^	L				X		
**ToMMV**	1	*S. lycopersicum* ^d^	L				X		
**ToMV**	1	*S. lycopersicum* ^d^	L				X		
**TPMVd**	1	*S. lycopersicum* ^d^	L				X		

^a^ Samples collected in Ortanova/Candela (FG) (Apulia region, Southern Italy) in 2022; ^b^ samples collected in Sperlonga/Fondi (LT) (Latium region, Central Italy) in 2022; ^c^ samples collected in Ortanova (FG) (Apulia region, Southern Italy) in 2023; ^d^ samples from CREA-DC collection. CEVd: citrus excortis viroid; CLVd: columnea latent viroid; CMV: cucumber mosaic virus; INSV: impatiens necrotic spot virus; PCFVd: pepper chat fruit viroid; PePMV: pepino mosaic virus; PmoV: parietaria mottle virus; PSTVd: potato spindle tuber viroid; PVY: potato virus Y; TASVd: tomato apical stunt viroid; TCDVd: tomato chlorotic dwarf viroid; TICV: tomato infectious chlorosis virus; TMV: tobacco mosaic virus; ToBRFV: tomato brown rugose fruit virus; ToCV: tomato chlorosis virus; ToLCNDV: tomato leaf curl New Delhi virus; ToMMV: tomato mild mottle virus; ToMV: tomato mosaic virus; TPMVd: tomato planta macho viroid.

**Table 2 viruses-16-00806-t002:** List of the mite samples used (X) in the set-up and validation of the real-time RT-PCR and ddRT-PCR assays for the identification of TRM at species level. The panel includes samples of TRM grown on healthy and ToFBV-infected *S. lycopersicum,* with non-target mite and insect species.

Arthropod Samples			Test Set-Up	Performance Criteria of Validation (EPPO PM7/98(5))			
Species	N° of Tested Samples	N. of Specimens/Sample		Analytical Sensitivity	Analytical Specificity (Inclusivity)	Analytical Specificity (Exclusivity)	Repeatability and Reproducibility
TRM ^a^	2	10	X				
TRM ^a^	2	5	X				
TRM ^a^	2	1	X				
TRM ^a^	5	10		X			
TRM ^a^	5	5		X			
TRM ^a^	5	3		X	X		
TRM ^a^	10	1		X			
TRM ^b^ from ToBFV-positive *S. lycopersicum* sample 1B	5	3			X		
TRM ^b^ from ToBFV-positive *S. lycopersicum* sample 1C	1	3			X		
TRM ^b^ from *S. nigrum*	6	3			X		
TRM ^b^ from *S. nigrum*	1	3			X		
*Aceria tosichella* ^c^	5	5				X	
*Phyllocoptes adalius* ^c^	5	5				X	
*B. tabaci* ^a^	1					X	
TRM ^a^	11	3	X				X
TRM ^a^	1	24					X

^a^ Samples from rearings at CREA-DC; ^b^ samples collected in Ortanova/Candela (FG—Apulia region, Southern Italy) in 2023; ^c^ samples from rearings at the Department of Plant Protection, Warsaw University of Life Sciences.

**Table 3 viruses-16-00806-t003:** Primer and probe sequences for the amplification of ToFBV (putative CP gene, ORF3 of RNA3) and TRM 28S rDNA (D2 region), and their positions on the sequences retrieved from GenBank.

Name	Sequence (5′–3′)	Position (Sequence ID)	Reference
ToFBV-Probe	FAM-TCCGAAATCCCGCCATCTTGTCAT-BH1	1879–1902 (NC_078394.1)	This study
ToFBV-F	CTCGTGATGTTGCCCATTTG	1847–1866 (NC_078394.1)	
ToFBV-R	GGAATTGCAGAGTAGGGAGAAT	1928–1949 (NC_078394.1)	
AcL-Probe	HEX- TGCTGGCTATGCGGCTGGTGGACT -BH1	195–218 (MT652212.1)	This study
AcL-F	CTTAGGATTTCGGTCCTATGGTG	171–193 (MT652212.1)	
AcL-R	TGCGCATTTTGTGTCAAGTC	258–277 (MT652212.1)	

**Table 4 viruses-16-00806-t004:** Results of analytical sensitivity assay for both real-time RT-PCR and ddRT-PCR tests targeting putative ToFBVCP gene in ORF3-RNA3, considering both single and duplex assays. For real-time RT-PCR, the Cq mean values (±SD) of the technical replicates are reported. For the ddRT-PCR, the number of copies/µL (±SD) on the Poisson calculation are reported. The LOD concentration is highlighted in grey. Samples 99, 95, and 1 (Table 1) were used.

	Real-Time RT-PCR			ddRT-PCR			Duplex Real-Time RT-PCR		
D	99	95	1	99	95	1	99	95	1
10^0^	8.70 ± 0.52	10.92 ± 0.02	15.68 ± 0.43	NT	NT	NT	10.52 ± 0.20	9.85 ± 0.01	17.02 ± 0.05
10^−1^	13.06 ± 0.02	13.52 ± 0.44	19.28 ± 0.02	NT	NT	NT	14.48 ± 1.02	15.01 ± 0.00	19.92 ± 0.13
10^−2^	16.33 ± 0.16	17.38 ± 0.02	22.15 ± 0.05	NT	NT	NT	17.28 ± 0.21	18.20 ± 0.35	22.35 ± 0.29
10^−3^	21.36 ± 0.08	20.43 ± 0.17	24.79 ± 0.1	NT	NT	NT	21.27 ± 0.03	21.59 ± 0.23	26.06 ± 0.06
10^−4^	24.4 ± 0.18	23.25 ± 0.01	28.42 ± 0.04	4480 ± 269	1700 ± 99	3655 ± 148	25.84 ± 0.67	23.87 ± 0.01	29.9 ± 0.17
10^−5^	28.98 ± 0.01	26.51 ± 0.18	32.21 ± 0.66	505 ± 53	136 ± 26	168 ± 33	28.97 ± 0.47	27.46 ± 0.47	33.45 ± 0.84
10^−6^	31.40 ± 0.19	28.77 ± 0.02	36.07 ± 1.18	75 ± 21	30 ± 13	19.5 ± 11	31.92 ± 0.29	30.00 ± 0.35	36.21 ± 0.05
10^−7^	33.69 ± 0.45	33.07 ± 0.01	Und	13 ± 8	4.3 ± 4	0	34.81 ± 0.06	33.28 ± 0.06	Und
10^−8^	36.80 ± 0.4	Und	Und	0	0	0	Und	Und	Und
10^−9^	Und	Und	Und	0	0	0	Und	Und	Und

D—dilutions; NT—not tested sample; Und—undetermined samples.

**Table 5 viruses-16-00806-t005:** Results of analytical sensitivity assay for both real-time RT-PCR and ddRT-PCR tests for the identification of TRM at the species level, considering both single and duplex assays. Five pools of ten, five, and three specimens and ten single specimens of TRM were tested. For the real-time RT-PCR, the minimum and maximum Cq values are reported. For the ddRT-PCR, the minimum and maximum number of copies/µL for each pool/specimen are reported. The LOD is highlighted in grey. Und—undetermined samples.

	n° Specimen/ Sample	n° Sample	Real-Time RT-PCR		ddRT-PCR		Duplex Real-Time RT-PCR	
			Min	Max	Min	Max	Min	Max
**TRM**	10	5	24.96	29.12	240	2530	24.40	27.87
**TRM**	5	5	26.63	30.45	116	723	25.46	29.42
**TRM**	3	5	28.31	34.05	8.9	275	29.03	35.25
**TRM**	1	10	34.38	Und	0	1.9	34.37	Und

## Data Availability

Dataset available on request from the authors.

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
