# Peer review of "Molecular Methods for the Simultaneous Detection of Tomato Fruit Blotch Virus and Identification of Tomato Russet Mite, a New Potential Virus–Vector System Threatening Solanaceous Crops Worldwide"

_viruses, 2024, doi:10.3390/v16050806_

Round 1
Reviewer 1 Report
Comments and Suggestions for Authors
Authors Marta Luigi and co-workers presented here a manuscript entitled “Molecular Methods for the Simultaneous Detection of Tomato Fruit Blotch Virus and Identification of Tomato Russet Mite, a New Potential Virus-Vector System Threatening Solanaceous Crops Worldwide”.
The main contribution of the paper is in the successful development of qRT-PCR and RT-ddPCR methods for simultaneous diagnosis of Tomato fruit blotch virus (ToFBV) (genus Blunervirus, family Kitaviridae) and its potential vector, the eriophyid mite Aculops lycopersici. Specific mite diagnosis is difficult and the authors describe that they have succeeded.
The optimization of the reaction for virus detection, mite identification and for the duplex reaction is described in detail. Validation of the assays for analytical sensitivity, analytical specificity, selectivity, repeatability and reproducibility is also described.
However, the table with primer and probe sequences is missing from the manuscript. The composition of the reactions is also missing. I cannot recommend the publication without those informations.
Minor comments:
Line 93: more and more species – does this mean virus species?
Line 95: highly specific - highly specific for what?
Line 129: Inpatiens - Impatiens
Line 132: Tomato infection chlorosis virus - Tomato infectious chlorosis virus
Line 158 and following: do not use abbreviation TRNA for total RNA. TRNA resembles tRNA, which is transfer RNA, and can be misleading.
Line 183: Koln, Germany – should be Köln
In Materials and Methods, I expect to see a description of the reaction composition, optimization and validation procedures.
Line 209: copies per l - should be a microliter
Comments on the Quality of English Language
English needs revision, there are strange sentence constructions and sometimes missing articles in the manuscript.
Author Response
Reviewer 1
Authors Marta Luigi and co-workers presented here a manuscript entitled “Molecular Methods for the Simultaneous Detection of Tomato Fruit Blotch Virus and Identification of Tomato Russet Mite, a New Potential Virus-Vector System Threatening Solanaceous Crops Worldwide”.
The main contribution of the paper is in the successful development of qRT-PCR and RT-ddPCR methods for simultaneous diagnosis of Tomato fruit blotch virus (ToFBV) (genus Blunervirus, family Kitaviridae) and its potential vector, the eriophyid mite Aculops lycopersici. Specific mite diagnosis is difficult and the authors describe that they have succeeded.
The optimization of the reaction for virus detection, mite identification and for the duplex reaction is described in detail. Validation of the assays for analytical sensitivity, analytical specificity, selectivity, repeatability and reproducibility is also described.
However, the table with primer and probe sequences is missing from the manuscript. The composition of the reactions is also missing. I cannot recommend the publication without those informations.
R: We completely agree with reviewer’s comment. During the transfer into Viruses word template the table 3 and part of the rest of M&M were erroneously omitted. In that table are included all information regarding primer/probe sequences. We apologize for this omission.
Minor comments:
Line 93: more and more species – does this mean virus species?
R: virus species, the text was amended
Line 95: highly specific - highly specific for what?
R: usually ETVs has a specific interaction with one species of mite as vector. We amended the text to make it clear
Line 129: Inpatiens – Impatiens
R: the text was amended
Line 132: Tomato infection chlorosis virus - Tomato infectious chlorosis virus
R: the text was amended
Line 158 and following: do not use abbreviation TRNA for total RNA. TRNA resembles tRNA, which is transfer RNA, and can be misleading.
R: we kindly do not agree with reviewer’s opinion. The acronym TRNA is usually accepted in the plant virology field.
Line 183: Koln, Germany – should be Köln
R: the text was amended
In Materials and Methods, I expect to see a description of the reaction composition, optimization and validation procedures.
R: as previously mentioned, part of the M&M was erroneously omitted. All information has been included in the current version of the manuscript.
Line 209: copies per l - should be a microliter
R: the text was amended
Reviewer 2 Report
Comments and Suggestions for Authors
The paper by Luigi et al. describes the development of molecular methods based on qPCR and ddPCR for the detection of an emergent kitavirus affecting tomatoes in Europe and other countries, and its tentative vector, the eriophyid mite Aculops lycopersici. The ms. deserves to be published because I consider it of scientific and public concern (tomato crop is economically important in many countries). However, the ms. is not ready to be published in its current form. The M&M section lacks a description of the development of the diagnostic tests. Besides, several minor recommendations/suggestions, and amendments are listed below. They need to be carefully addressed by the authors.
Line 31: Italicize the viral species name
Line 58: “….is reported in samples….” Add the missing preposition.
Lines 67-69: The indication of the GenBank accession number is likely a more useful way to refer to these findings.
Lines 82-83: A reference to support the statement and provide a wider and more holistic knowledge of the family Kitaviridae may be necessary. A recent review is available: https://doi.org/10.1146/annurev-phyto-021622-121351
Line 84-85: hibiscus green spot virus “2” and citrus leprosis virus “C”. Add “2” and “C”, respectively, to agree with the formal name/designation of these viruses. Moreover, it is strongly recommended to add the binomial names of the species these two viruses belong.
Lines 86-87: “….. nonetheless, the other two classified blunerviruses are transmitted by mites in the family Eriophyidae”. This phrase seems inaccurate. Vector-mediated transmission of blunerviruses has been demonstrated yet. I recommend toning down the statement by suggesting instead of confirming. This subject may have been speculated, but, indeed, accurate transmission experiments have been not carried out. Significant results have been obtained for BNRBV, and the papers is coming, according to the information released at https://doi.org/10.1146/annurev-phyto-021622-121351.
Lines 93-94: “Numerous eriophyid species have economic significance for their primary damaging action [15]; however, in the last decade, more and more species are characterized as Eriophyid transmitted viruses (ETVs).” Are you referring to mites or viruses? Please, reword.
Line 96: Could you format the reference “Stenger et al. 2016”.
Lines 154-155: Letters “b and c” in the table’s footnote must be reformatted and positioned as superscripts.
Line 162: PO4 can be removed.
Line 171: What ORF in which of the viral genomic RNA molecules is expected to encode the coat protein? Add a reference to support that statement. By the way, in the legend of Table 4, “…targeting ToFBV-CP gene,…..” and in line 444 “targeted the CP gene”. Altogether these phrases seem to be contradictory.
Line 183: Köln?
Line 184: The M&M section seems to be truncated. The description of qPCR and ddPCR development is missing. Add the methodology corresponding to the choice of the optimal annealing temperature and primer and probe concentrations, number of technical replicates, etc. Details of for example: Repeatability and reproducibility experiments, need to be informed here. Observe that some of this information is included in the figure legends, which is not incorrect at all because the figure needs to be self-explanatory.
Line 204: Describe what the horizontal blue line in each graphic means. The threshold lines?
Line 209: “target copies per l was” Something is missing.
Figure 2: Is the X-axis of Figure 2a representing samples or the annealing temperature?
Line 264: Table 3 is missing.
Table 4: Indicate what the column headings 99, 95, and 1 mean. sample ID??
Line 288: To identify or to detect???
Lines 321 and 327. It is recommended to add these non-shown results in a supplementary table.
Lines 368-374: Some of the information here is typical of M&M section. Please, rewrite.
Line 371: “….different instruments.” What instruments??
Line 377: “….blue and green, respectively”
Figure 10: What colors stand for? Agilent???
Lines 491-492: “For ToFBV, this allows to measure the absolute number of viral RNA copies in the TRNA sample without ….” This seems not to be accurate. The ORF target of detection assays described in this study is likely expressed from a subgenomic viral RNA, in that case, the output is a sum of viral genome + subgenomic RNA.

Author Response
Reviewer 2
The paper by Luigi et al. describes the development of molecular methods based on qPCR and ddPCR for the detection of an emergent kitavirus affecting tomatoes in Europe and other countries, and its tentative vector, the eriophyid mite Aculops lycopersici. The ms. deserves to be published because I consider it of scientific and public concern (tomato crop is economically important in many countries). However, the ms. is not ready to be published in its current form. The M&M section lacks a description of the development of the diagnostic tests. Besides, several minor recommendations/suggestions, and amendments are listed below. They need to be carefully addressed by the authors.
We completely agree with reviewer’s comment. During the transfer into Viruses word template the table 3 and part of the rest of M&M were erroneously omitted. In that table are included all information regarding primer/probe sequences. We apologize for this omission
Line 31: Italicize the viral species name
R: the text was amended
Line 58: “….is reported in samples….” Add the missing preposition.
R: the text was amended
Lines 67-69: The indication of the GenBank accession number is likely a more useful way to refer to these findings.
R: the text was amended
Lines 82-83: A reference to support the statement and provide a wider and more holistic knowledge of the family Kitaviridae may be necessary. A recent review is available: https://doi.org/10.1146/annurev-phyto-021622-121351
R: the text was amended
Line 84-85: hibiscus green spot virus “2” and citrus leprosis virus “C”. Add “2” and “C”, respectively, to agree with the formal name/designation of these viruses. Moreover, it is strongly recommended to add the binomial names of the species these two viruses belong.
R: the text was amended
Lines 86-87: “….. nonetheless, the other two classified blunerviruses are transmitted by mites in the family Eriophyidae”. This phrase seems inaccurate. Vector-mediated transmission of blunerviruses has been demonstrated yet. I recommend toning down the statement by suggesting instead of confirming. This subject may have been speculated, but, indeed, accurate transmission experiments have been not carried out. Significant results have been obtained for BNRBV, and the papers is coming, according to the information released at https://doi.org/10.1146/annurev-phyto-021622-121351.
R: the text was amended
Lines 93-94: “Numerous eriophyid species have economic significance for their primary damaging action [15]; however, in the last decade, more and more species are characterized as Eriophyid transmitted viruses (ETVs).” Are you referring to mites or viruses? Please, reword.
R: the text was amended
Line 96: Could you format the reference “Stenger et al. 2016”.
R: the text was amended
Lines 154-155: Letters “b and c” in the table’s footnote must be reformatted and positioned as superscripts.
R: the text was amended
Line 162: PO4 can be removed.
R: we would like to maintain the acronym since it is cited in those parts of M&M that have been included in the current version of the manuscript (lines 300, 324 etc…).
Line 171: What ORF in which of the viral genomic RNA molecules is expected to encode the coat protein? Add a reference to support that statement. By the way, in the legend of Table 4, “…targeting ToFBV-CP gene,…..” and in line 444 “targeted the CP gene”. Altogether these phrases seem to be contradictory.
R: the text was amended
Line 183: Köln?
R: the text was amended
Line 184: The M&M section seems to be truncated. The description of qPCR and ddPCR development is missing. Add the methodology corresponding to the choice of the optimal annealing temperature and primer and probe concentrations, number of technical replicates, etc. Details of for example: Repeatability and reproducibility experiments, need to be informed here. Observe that some of this information is included in the figure legends, which is not incorrect at all because the figure needs to be self-explanatory.
R: as previously mentioned, part of the M&M was erroneously omitted. All information has been included in the current version of the manuscript
Line 204: Describe what the horizontal blue line in each graphic means. The threshold lines?
R: the text was amended
Line 209: “target copies per l was” Something is missing.
R: the text was amended
Figure 2: Is the X-axis of Figure 2a representing samples or the annealing temperature?
R: the annealing temperature. As reported in the caption of the figure, the same sample (loaded eight times) was amplified using 8 different annealing temperatures.
Line 264: Table 3 is missing.
R: the text was amended
Table 4: Indicate what the column headings 99, 95, and 1 mean. sample ID??
R: the text was amended
Line 288: To identify or to detect???
R: identify
Lines 321 and 327. It is recommended to add these non-shown results in a supplementary table.
R: A supplementary table was included
Lines 368-374: Some of the information here is typical of M&M section. Please, rewrite.
R: the text was amended
Line 371: “….different instruments.” What instruments??
R: the text was amended
Line 377: “….blue and green, respectively”
R: the text was amended
Figure 10: What colors stand for? Agilent???
R: We hope the experiments can be correctly interpreted after adding the missing part of M&M.
Lines 491-492: “For ToFBV, this allows to measure the absolute number of viral RNA copies in the TRNA sample without ….” This seems not to be accurate. The ORF target of detection assays described in this study is likely expressed from a subgenomic viral RNA, in that case, the output is a sum of viral genome + subgenomic RNA.
R: the text was amended
Reviewer 3 Report
Comments and Suggestions for Authors
While reading some questions arise, which are (rather well) addressed in discussion section and later become clear. But while reading they constantly pester the reader. Thus, I recommend shortly mentioning them in the last paragraph of introduction section. Mainly formulating better, the aim and necessity of your work:
1) why RNA approach was selected for mite identification (higher concentration than DNA, single approach for two analyses, single nucleic acid isolation from leaves…).
2) why mite morphological identification is tricky (because they are microscopic…)
3) also, why complicated and expensive qPCR and digital/droplet PCR were chosen if the priority is simple infection tests for quarantine/import of plant material? Is speed so important compared to conventional RT-PCR? It is not that much faster. And the cost is more than double. Also, I would argue that digital PCR also requires “extensive skills and expertise” from staff. Conventional PCR sensitivity to low concentrations can be increased by nested PCR or doing 2 step RT-PCR (synthesizing cDNA in separate reaction).
These things should be mentioned in the beginning of your article so the reader fully understands the necessity of your work and why you made certain choices before he gets to the end of the article. Longer explanations in discussion are fine.
Line 17: First mention of HTS, I recommend adding full text naming.
Line 56: “countries and continents” sound weird. Maybe “other continents” or list the particular continents.
Line 181: The table 3 is missing!
Author Response
Reviewer 3
While reading some questions arise, which are (rather well) addressed in discussion section and later become clear. But while reading they constantly pester the reader. Thus, I recommend shortly mentioning them in the last paragraph of introduction section. Mainly formulating better, the aim and necessity of your work:
R: We completely agree with reviewer’s comment. We amended the manuscript accordingly. In addition, during the transfer into Viruses word template the table 3 and part of the rest of M&M were erroneously omitted. In that table are included all information regarding primer/probe sequences. We apologize for this omission
- why RNA approach was selected for mite identification (higher concentration than DNA, single approach for two analyses, single nucleic acid isolation from leaves…).
R: the text was amended
- why mite morphological identification is tricky (because they are microscopic…)
R: the text was amended
- also, why complicated and expensive qPCR and digital/droplet PCR were chosen if the priority is simple infection tests for quarantine/import of plant material? Is speed so important compared to conventional RT-PCR? It is not that much faster. And the cost is more than double. Also, I would argue that digital PCR also requires “extensive skills and expertise” from staff. Conventional PCR sensitivity to low concentrations can be increased by nested PCR or doing 2 step RT-PCR (synthesizing cDNA in separate reaction).
R: the text was amended
These things should be mentioned in the beginning of your article so the reader fully understands the necessity of your work and why you made certain choices before he gets to the end of the article. Longer explanations in discussion are fine.
Line 17: First mention of HTS, I recommend adding full text naming.
R: the text was amended
Line 56: “countries and continents” sound weird. Maybe “other continents” or list the particular continents.
R: the text was amended
Line 181: The table 3 is missing!
R: the text was amended
Round 2
Reviewer 1 Report
Comments and Suggestions for Authors
Authors Marta Luigi and co-workers presented here a revised version of a manuscript entitled “Molecular methods for the simultaneous detection of tomato fruit blotch virus and identification of Tomato Russet Mite, a new potential virus-vector system threatening solanaceous crops worldwide”.
The main contribution of the paper is the optimisation and validation of the qRT-PCR detection of the virus and its vector mite.
The authors have added missing parts to the manuscript and corrected minor errors. They have addressed all my comments. The manuscript is now ready for publication.